# Anti-GD2 induced allodynia in rats can be reduced by pretreatment with DFMO

**Mitchell B. Diccianni**[ID][1]*, **Katarzyna Kempińska**[1], **Jon A. Gangoti**[1], **Alice L. Yu**[1,2], **Linda S. Sorkin**[3]

**1** Department of Pediatrics, University of California San Diego, San Diego, California, United States of America, **2** Institute of Stem Cell and Translational Cancer Research, Chang Gung Memorial Hospital at Linkou & Chang Gung University, Taoyuan City, Taiwan, **3** Department of Anesthesiology, University of California San Diego, San Diego, California, United States of America

\* mdiccianni@ucsd.edu

## Abstract

### Background

Anti-GD2 therapy with dinutuximab is effective in improving the survival of high-risk neuroblastoma patients in remission and after relapse. However, allodynia is the major dose-limiting side effect, hindering its use for neuroblastoma patients at higher doses and for other GD2-expressing malignancies. As polyamines can enhance neuronal sensitization, including development of allodynia and other forms of pathological pain, we hypothesized that polyamine depletion might prove an effective strategy for relief of anti-GD2 induced allodynia.

### Method

Sprague-Dawley rats were allowed to drink water containing various concentrations of difluoromethylornithine (DFMO) for several days prior to behavioral testing. Anti-GD2 (14G2a) was injected into the tail vein of lightly sedated animals and basal mechanical hindpaw withdrawal threshold assessed by von Frey filaments. Endpoint serum DFMO and polyamines, assessed 24h after 14G2a injection, were measured by HPLC and mass spectrometry.

### Results

An i.v. injection of 14G2a causes increased paw sensitivity to light touch in this model, a response that closely mimics patient allodynia. Animals allowed to drink water containing 1% DFMO exhibited a significant reduction of 14G2a-induced pain sensitivity (allodynia). Increasing the dosage of the immunotherapeutic increased the magnitude (intensity and duration) of the pain behavior. Administration of DFMO attenuated the enhanced sensitivity. Consistent with the known actions of DFMO on ornithine decarboxylase (ODC), serum putrescene and spermidine levels were significantly reduced by DFMO, though the decrease in endpoint polyamine levels did not directly correlate with the behavioral changes.

**Data Availability Statement:** All relevant data are within the paper and its Supporting Information files.

**Funding:** This study was funded by United Therapeutics Corporation. The funders had no role

in study design, data collection and analysis, decision to publish, or preparation of the manuscript.

**Competing interests:** This study was funded by United Therapeutics Corporation. The funders had no role in study design, data collection and analysis, decision to publish, or preparation of the manuscript. This does not alter our adherence to PLOS ONE policies on sharing data and materials.

## Conclusions

Our results demonstrate that DFMO is an effective agent for reducing anti-GD2 -induced allodynia. Using DFMO in conjunction with dinutuximab may allow for dose escalation in neuroblastoma patients. The reduction in pain may be sufficient to allow new patient populations to utilize this therapy given the more acceptable side effect profile. Thus, DFMO may be an important adjunct to anti-GD2 immunotherapy in addition to a role as a potential anti-cancer therapeutic.

## Introduction

GD2 is a disialoganglioside found on the outer cell membrane and is believed to play a role in neuronal development, differentiation and repair [1]. Prenatal expression of GD2 is found principally on neural and mesenchymal stem cells, with postnatal expression limited to peripheral nerves, elements of the central nervous system, and skin melanocytes [2]. Importantly, many cancer cells including neuroblastoma express GD2 on their surface [3]. Until recently, approximately two-thirds of patients diagnosed with high risk neuroblastoma would succumb to the disease despite obtaining remission. The abundant expression of GD2 on neuroblastoma but limited expression on normal cells made it an attractive target for anti-GD2 immunotherapy. We have reported that anti-GD2 (dinutuximab) is efficacious in improving neuroblastoma patient survival when administered to patients in remission as well as in relapsed or refractory disease [4, 5]. However, late relapses that diminish overall survival do occur [6, 7]. Although an increase in dosage or number of cycles of dinutuximab could potentially reduce late relapses, this approach is hampered by an increase in dinutuximab-associated toxicities. In particular, whole body allodynia, which is severe pain perceived in response to light touch, is the major side effect of dinutuximab, limiting its expanded usage and dosage. To address this problem, co-administration of morphine or other narcotics is common. Despite such measures, some patients still experience severe pain that interferes with the activities of daily living or totally disabling pain [4]. Most toxicities can be reduced, in part, by increasing infusion duration while maintaining overall dosage [8]. However, allodynia remains the major and the dose-limiting toxicity even on this modified schedule.

High levels of polyamines and ornithine decarboxylase (ODC) activity, the rate limiting enzyme in polyamine biosynthesis, are found in many human cancers including neuroblastoma [9, 10]. Mammalian cells sequentially synthesize three polyamines from ornithine. The first product is putrescine, which is then converted into spermidine and spermine. Difluoromethylornithine (DFMO, eflornithine) is an inhibitor of ODC which reduces serum polyamine levels with minimal toxicity but has little stand-alone anti-cancer activity [11]. On the other hand, DFMO in combination with various anti-cancer agents with diverse mechanisms of action have shown promising results in clinical trials [12–15].

Polyamines have also been linked to the nociceptive pathway. Polyamines can induce neuronal sensitization as well as the development of allodynia and hyperalgesia [16]. Consistent with this, a reduction in polyamine levels significantly reduced inflammation-induced and neuropathic pain in animal models [17]. In other pre-clinical studies, rats fed a polyamine deficient diet (PDD) displayed significantly less oxaliplatin-induced pain behavior [18]. In prostate cancer patients, a PDD decreased patient-reported cancer-associated pain with no detrimental side effects [19]. These results suggest that decreasing polyamines can reduce both cancer associated pain as well as pain arising from other sources.

The observation that for a variety of cancers and iatrogenic pain states a reduction in poly-amine levels may improve quality of life has led us to hypothesize that polyamine depletion might also be an effective anti-allodynia strategy for anti-GD2 immunotherapy. Thus, this study had two objectives 1) to test the hypothesis that DFMO would alleviate anti-GD2 induced allodynia and 2) to determine if there is an association between plasma levels of poly-amines and magnitude of pain behavior. The first objective was considered to be of primary clinical importance. Using a rat model of pain behavior, oral DFMO significantly reduced the allodynia induced by an i.v. injection with 14G2a, the murine version of dinutuximab. Our results suggest that DFMO may be an effective agent for reducing anti-GD2-induced allodynia and may be an important adjunct to dinutuximab immunotherapy in addition to, but regard-less of, any function as an anti-cancer therapy adjunct.

## Methods

### Animals and experimental design

Experimental protocols were approved by the Institutional Animal Care and Use Committee of the University of California, San Diego. (Protocol Number: S12314). All animal research fol-lowed the ARRIVE guidelines (S1 Appendix). Sixty-six male Sprague-Dawley rats (Envigo, Indianapolis, IN) weighing 200-250g were housed in pairs and kept on a 12-h light/dark cycle. Food and water were available *ad libitum*. Seven to ten days prior to the experiment, drinking water was replaced with an aqueous solution containing 0.25%, 0.5% or 1.0% DMFO or water alone. The DFMO solutions were visually identical to plain water. Animals were weighed after arrival and then again at the start of the experiment. Animals were qualitatively assessed for any ill effects of the DFMO on motor skills, reflexes and general health activity to look for pos-sible DFMO behavioral side effects; none were observed.

For three days prior to the experiment, animals were acclimated to the behavioral testing room, equipment and procedures. On the day of the experiment, animals were placed in indi-vidual plastic test chambers with wire mesh floors to allow access to the hindpaws. Basal mechanical hindpaw withdrawal threshold (50% probability) was assessed by means of a set of calibrated von Frey filaments (Stoelting, Wood Dale, IL) with buckling forces between 0.41–15.1g. When the animal was quiet and resting on all four paws, a filament, beginning with the 2.0g filament, was pressed perpendicularly against the surface of the hindpaw until it bent slightly. A response consisted of either a brisk movement away from the probe (escape) or a paw lick. Pressure was maintained for 6s or until a response occurred. Successive stimuli were separated by 15 to 30 seconds. Stimuli were presented in ascending order of stiffness until a response occurred or the stiffest filament in the set was used, statistically these animals were treated as having a 15.1g response. If a response occurred, filaments of decreasing strength were applied until the animal no longer responded, at which point filaments were again presented in ascending order. This pattern was repeated for four stimuli presentations after the first with-drawal response, and the 50% probability withdrawal threshold determined [20]. Following baseline determination, rats were lightly anesthetized with 2% isoflurane and 0, 1 or 2 mg/kg of 142Ga in saline injected via the tail vein. Rats were immediately returned to the test chamber and thresholds re-determined at 0.5, 1, 2, 3, 4, 5, 8 and 24 h post injection. Rats were returned to their home cages for the periods after the 5 and 8 h time points to allow them access to food and fluids. All experiments began between 9 and 10 a.m. The person performing the behavioral testing was unaware of the contents of the fluid in the water bottles. At the end of experiments, the rats were deeply anesthetized with 5% isoflurane and blood withdrawn via cardiac puncture for polyamine and DFMO analysis before sacrifice by $CO_2$ and bilateral pneumothorax in accordance with American Veterinary Medical Association (AVMA) Guidelines.

## DFMO and anti-GD2

Murine anti-GD2 (14G2a) was manufactured by BioTechnetics Inc. San Diego CA and was from clinical stock originally used in Phase I trials for neuroblastoma [21]. The morning of the experiment it was diluted in physiological saline and administered at a dosage of 1 or 2 mg/kg. DFMO was synthesized by Genzyme Corporation (Lot #PD39CF) and graciously provided by Professor Patrick M. Woster, Ph.D. from Medical University of South Carolina, Charleston, South Carolina. It was diluted in tap water on the day that it was first used and placed in the animals' water bottles at room temperature until the end of the experiment. Bottles were refilled as necessary.

## Polyamine and DFMO measurement

If the first objective was met, our second objective was to determine the relationship with serum levels of polyamines, which are known to be reduced by oral DFMO, with magnitude of pain behavior. Serum was stored at -20°C before being batch sent for DFMO and polyamine analysis at the UCSD Biochemical Genetics and Metabolomics Laboratory (http://ucsdbglab. org/metabolomics/Panels.asp) using high performance liquid chromatography and mass spectrometry (LC-MS/MS). Briefly, 100μL of rat serum was vortex-mixed with 1mL of ice-cold methanol containing stable-isotope labeled internal standards ($^2$H$_8$-putrescine (d$_8$-putrescine), $^2$H$_{20}$-spermine (d$_{20}$-spermine) and $^2$H$_2$-ornithine, incubated for 30 minutes at -20°C and centrifuged at 17,136 x $g$ at 5°C. Supernatants were evaporated to dryness and reconstituted in 50μL of 5% acetonitrile + 3.6mM ammonium formate+0.1% formic. Polyamines and DFMO were separated at room temperature on a 150 x 2.1mm (5μm particle size) TSKgel Amide-80 column at a flow rate of 0.3mL/min. Identities of polyamines and DFMO were validated by LC-MS/MS.

Because of the inability to reliably measure spermidine or spermine using our initial methodology, we developed a second method to measure polyamines adapted from [22] and [23]. Urethane derivatives di-isobutoxy- ($^i$BuO) putrescine, tri- $^i$BuO-spermidine and tetra- $^i$BuO-spermine were analyzed using above described stable-isotope dilutions, with di- $^i$BuO-d$_8$-putrescine and tetra-$^i$BuO-d$_{20}$-spermine internal standards. Briefly, rat serum was deproteinized with 4% trichloracetic acid, adjusted to pH 9 with ammonium hydroxide and ammonium formate buffer, and reacted with isobutyl chloroformate at 35°C. Carbamoyl derivatives were extracted by solid-phase extraction (SPE) on a 30mg Strata-X reversed phase cartridge conditioned with methanol and water, washed with 5% acetonitrile, eluted with 90% acetonitrile + 0.1% formic acid and dried. Eluate was then reconstituted in 50% acetonitrile + 0.1% formic acid and analyzed in gradient mode by LC-MS/MS on a Sciex API4000 MS coupled to an Agilent 1200 series LC on a Kinetex C18 LC column (100x2.1mm; 5μm).

Further details on polyamine analysis methodology and raw data on the figures presented in this manuscript are available as supplemental information (S1 and S2 Methods and S1 Table and S1 Raw data).

## Statistics

Sample size determination for allodynia experiments was determined at 80% power and a significance of 0.05 using the standard sample size calculations [24] and using the maximum values of allodynia determined for 1 mg/ml 14G2a that we have reported [25]. Groups were compared by one-way ANOVA with post hoc analyses by Fishers LSD or repeated measures ANOVA with post hoc analyses by Bonferroni test. Two groups were compared using t-test, or t-test with Welch correction when variances were significantly different. We also examined differences in pain behavior among groups by calculating the area under the curve (AUC,

hyperalgesia index) for each animal, which compresses the data for the entire time course for into a single data point. Paw withdrawal response (pain behavior data), polyamine and DFMO concentration are presented as the mean and standard error of all observed values at each timepoint. Serum polyamine levels were derived from duplicate measurements of individual samples. Data were analyzed using OriginPro 2018 and Graphpad Prism; $p < 0.05$ was considered statistically significant.

## Results

### Anti-GD2 treatment induces a dose dependent allodynia that can be reduced by DFMO

We have previously reported that an intravenous injection of 1 mg/kg 14G2a, a dosage within the range used in children, into rat tail vein induces withdrawal responses at pressures that are normally innocuous, indicative of pain behavior (allodynia) [25]. Consistent with this finding, animals injected with both 1 mg/kg (N = 14) and 2 mg/kg (N = 6) 14G2a developed significant allodynia with similar onsets of development (latency) within 1-hour post i.v. injection, and with maximum allodynia occurring about 3-5hs post-injection (Fig 1A and 1C, $p < 0.001$, one-way repeated measures ANOVA). For the 1 mg/kg dose of 14G2a, thresholds remained significantly lower than pre-injection levels at all time points through 8h while by 24h, it was no longer different from the pre-injection control values (Fig 1A and Table 1). In contrast, sensitivity

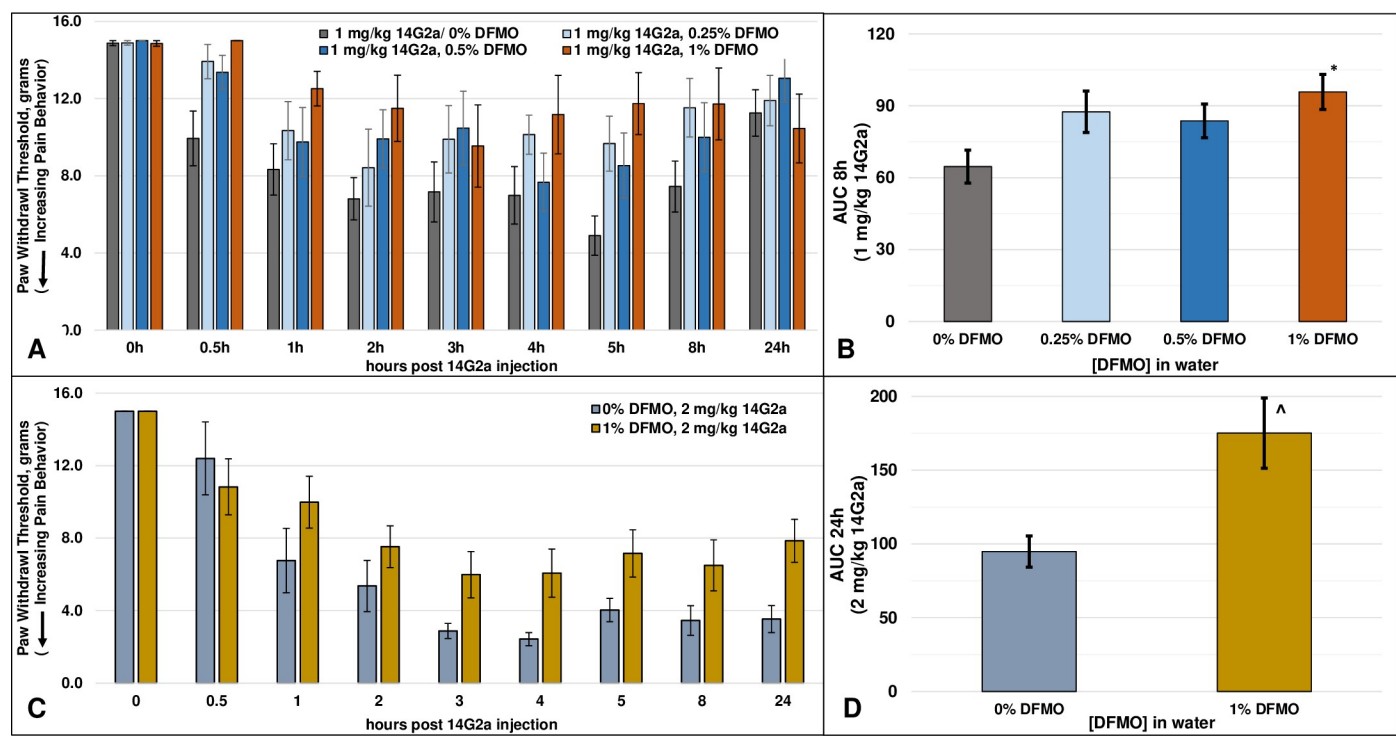

**Fig 1. Rat paw withdrawal responses after injection with anti-GD2. A)** Rat paw withdrawal responses (allodynia, sensitivity) over the course of the study in animals drinking water without DFMO after a 1 mg/kg injection of 14G2a ($p < 0.001$, one-way Repeated Measures ANOVA), in animals drinking water containing 0.25% and 0.5% DFMO (both $p < 0.005$, one-way Repeated Measures ANOVA), and in animals drinking water containing 1% DFMO ($p > 0.05$, one-way Repeated Measures ANOVA); **B)** The AUC after a 1 mg/kg injection of 14G2a and each DFMO concentration for the first 8h (AUC 8h) ($p = 0.077$, One-way ANOVA) (*$p < 0.05$, 0% DFMO vs. 1% DFMO, post-hoc Fishers LSD test); **C)** Rat paw withdrawal responses in animals injected with 2 mg/kg injection of 14G2a and allowed to drink water without or with 1% DFMO (both conditions versus pre-treatment, $p < 0.001$, one-way Repeated Measures ANOVA); **D)** The AUC after a 2 mg/kg injection of 14G2a for the entire 24h (AUC 24h) (^$p < 0.01$, t-test).

**Table 1. Bonferroni comparisons vs pre-injections.**

| Time | 1 mg 14G2a | | | | 2 mg 14G2a | |
|---|---|---|---|---|---|---|
| | 0% DFMO | 0.25% DFMO | 0.5% DFMO | 1% DFMO | 0% DFMO | 1% DFMO |
| 0h vs 0.5h | p<0.05 | p>0.05 | p>0.05 | p>0.05 | p>0.05 | p>0.05 |
| 0h vs 1h | p<0.001 | p>0.05 | p>0.05 | p>0.05 | p<0.001 | p<0.05 |
| 0h vs 2h | p<0.001 | p<0.001 | p>0.05 | p>0.05 | p<0.001 | p<0.001 |
| 0h vs 3h | p<0.001 | p<0.05 | p>0.05 | p>0.05 | p<0.001 | p<0.001 |
| 0h vs 4h | p<0.001 | p<0.05 | p<0.05 | p>0.05 | p<0.001 | p<0.001 |
| 0h vs 5h | p<0.001 | p<0.05 | p<0.05 | p>0.05 | p<0.001 | p<0.001 |
| 0h vs 8h | p<0.001 | p>0.05 | p>0.05 | p>0.05 | p<0.001 | p<0.001 |
| 0h vs 24h | p>0.05 | p>0.05 | p>0.05 | p>0.05 | p<0.001 | p<0.001 |

Post-hoc Bonferroni p values following one-way repeated measures ANOVA (see text and Fig 1) of each testing condition at each timepoint versus pretreatment.

to touch remained significantly different than basal levels even at 24h in the 2 mg/kg 14G2a treated animals, the latest time-point investigated, demonstrating a longer duration of effect (Fig 1C and Table 1). At every time point measured from 1h until the end of the study, a qualitative examination of the data indicates that animals given 2 mg/kg had lower average withdrawal thresholds (more allodynia) than those given the lower dose. A comparison of the hyperalgesia indexes of the two 14G2a dose groups in the absence of DFMO over both the first 8h (AUC8h: 68.0 ± 7.2 (1 mg/kg/0% DFMO, N = 14) vs 46.6 ± 3.7 (2 mg/kg/0% DFMO, N = 6), p<0.02, t-test), as well as the entire 24h experimental period (AUC24h: 216.9 ± 21.0 (1 mg/kg/0% DFMO, N = 14) vs. 94.8 ± 10.6 (2 mg/kg/0% DFMO, N = 6), p<0.001, t-test), confirms the significantly greater allodynia at the higher 14G2a dosage. These data demonstrate a dose-dependent increase in pain behavior elicited by 14G2a, whereas an increased dosage of 14G2a increased the magnitude and duration of the pain behavior without affecting the latency of onset.

Prior to comparing allodynic profiles of rats given anti-GD2 and DFMO, we first established that pretreatment with DFMO did not affect baseline withdrawal thresholds in otherwise untreated animals (T = 0h, Fig 1A and 1C). We next compared anti-GD2 induced pain behavior in animals administered water-containing various concentrations of DFMO. Animals in the 0.25% and 0.5% DFMO groups (N = 10 each) continued to exhibit significant pain behavior in response to injection with 1 mg/kg 14G2a (Fig 1A, 0.25% (p<0.001) and 0.5% (p<0.005), one-way repeated measures ANOVA). Although sensitivity persisted in animals' drinking water with 0.25% and 0.5% DFMO, pain-behavior was nevertheless reduced, particularly at the beginning and end of the experiment (Table 1). In animals given the higher concentration of 1% DFMO (N = 8), withdrawal thresholds were not significantly different from pre-injection levels at any timepoint (p>0.05, repeated measures ANOVA, Fig 1A and Table 1).

We also evaluated the effect of DFMO on allodynia over the entire experiment using an AUC analysis (Fig 1B). Based on the duration of the 14G2a-induced allodynia in the non-DFMO watered animals, we limited the AUC analysis of the 1 mg/kg dosage to the first 8h, the time period during which 14G2a produced significant allodynia. After normalizing the data in this manner, the differences among the four treatment groups did not reach statistical significance (p = 0.077, ANOVA), though post-hoc and t-test analyses both document that there was a significant difference between the control (non-DFMO) group and the 1% DFMO group (p<0.05). The lower concentrations of DFMO, which both reduced the magnitude and duration but did not fully abrogate the allodynia (Fig 1A and Table 1), failed to demonstrate significant differences from the non-DFMO group using this normalization approach (Fig 1B).

In animals allowed to drink water without (N = 6) or with (N = 12) 1% DFMO and injected with 2 mg/kg 14G2a, the antibody induced significant allodynia under both conditions (both p<0.001, one-way repeated measures ANOVA, Fig 1C). However, in contrast to the animals injected with 1 mg/kg 14G2a, 1% DFMO reduced but did not completely prevent pain behavior, with thresholds remaining significantly lower than pre-injection levels at all timepoints from 1h on (Table 1). As allodynia in this group remained significant even at the end of the experiment, we utilized the entire 24h time course for an AUC analysis. The significantly greater AUC in the 1% DFMO versus the non-DFMO group (p<0.01, t-test, Fig 1D) clearly demonstrates that there is a decrease in overall allodynia in the animals drinking water plus DFMO despite the increased pain burden in the control group.

These data demonstrate a dose-dependent increase in pain behavior induced by 14G2a, where an increased dosage of 14G2a increased the magnitude and duration of the pain behavior without significantly affecting the latency of onset. More importantly, these data demonstrate the ability of DFMO to reduce the magnitude and duration of significant pain behavior and imply that increased dosage of anti-GD2 treatment may be made tolerable in the presence of DFMO co-administration.

## Serum concentrations of DFMO

Animals were allowed to freely drink water containing different concentrations of DFMO (0.25%, 0.5% and 1%) or water alone. All animals, regardless of treatment, looked healthy without any evidence of lethargy and showed normal weight gain over the course of the study. There were no signs of dehydration in any of the groups. At the end of the study serum levels of DFMO were determined to compare how the administered concentrations were reflected in the blood of the treated animal. No HPLC peak for DFMO was seen in the control animals, as expected. In contrast, all animals given DFMO in their water had a positive HPLC peak, with mean ± SEM DFMO levels of 17. 8 +/- 5.1 µM (N = 10), 16.5 +/- 2.7 µM (N = 10) and 35.3 +/- 5.4 µM (N = 20) for the 0.25%, 0.5% and 1% groups respectively (Fig 2). ANOVA analysis of the three groups that received DFMO in their water demonstrated there were significant

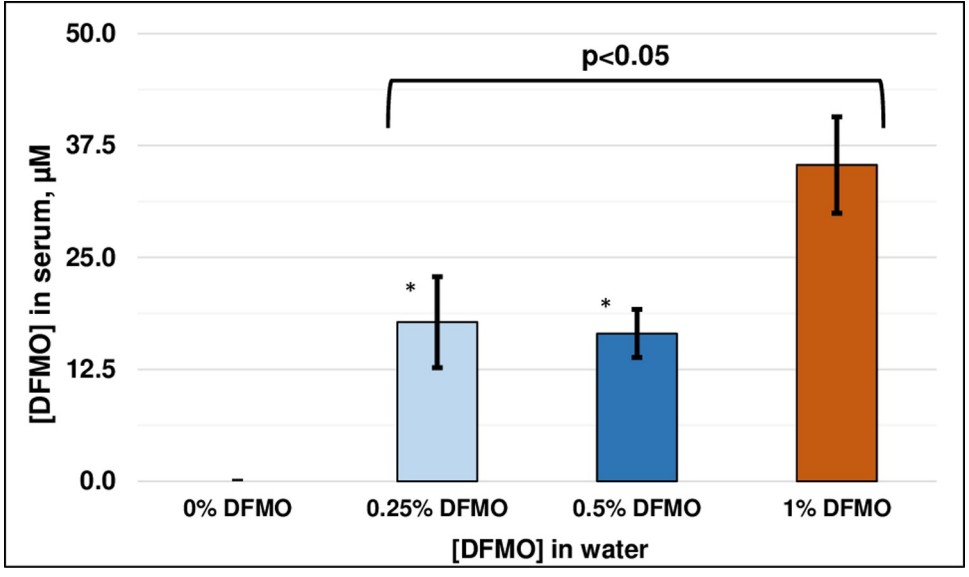

**Fig 2. DFMO levels in rat blood.** Serum DFMO levels at the end of the study (mean ± SEM). *p<0.05, 0.25% and 0.5% vs. 1% DFMO, post-hoc Fishers LSD test.

differences between the groups (p<0.05), with post-hoc comparisons indicating that differences were between either of the two lower DFMO concentrations and the 1% group (p<0.05 each), while the 0.25% and 0.5% DFMO groups were not different from each other.

## Changes in serum polyamines by DFMO

We compared polyamine levels in three untreated (non-DFMO watered, non-14G2a) animals versus non-DFMO watered animals injected with both doses of 14G2a. Though mean putrescene levels were slightly higher in animals injected with 2 mg/kg than 1 mg/kg, and mean spermidine and spermine levels were slightly higher in 14G2a treated versus untreated animals, a comparison of each of the three polyamines at untreated, 1 mg/kg and 2 mg/kg 14G2a revealed that none of these changes reached significance (Fig 3, ANOVA each polyamine, p>0.05).

As DFMO is known to reduce serum polyamine levels, and increased serum polyamines are observed in some clinical pain states and various manipulations aimed at reducing serum polyamines reduce pain behavior, we measured serum polyamines following both dosages of 14G2a and DFMO pretreatment. Consistent with the known actions of DFMO, serum putrescine levels were significantly reduced in the 0.25%, 0.5% and 1% groups of DFMO-treated animals following 1 mg/kg 14G2a compared to animals not given DFMO (Fig 3A, 0.68 μM ± 0.09 (0%, N = 14), 0.35 ± 0.04 μM (0.25%, N = 10), 0.31 ± 0.02 μM (0.5%, N = 10) and 0.29 ± 0.03μM (1%, N = 7); ANOVA p<0.001). Post-hoc analyses demonstrated the mean putrescine levels of each of the DFMO groups were significantly lower than the untreated group but not different from each other, suggesting levels are already maximally decreased at the lower concentrations of DFMO. Putrescine is the first polyamine synthesized by ODC and

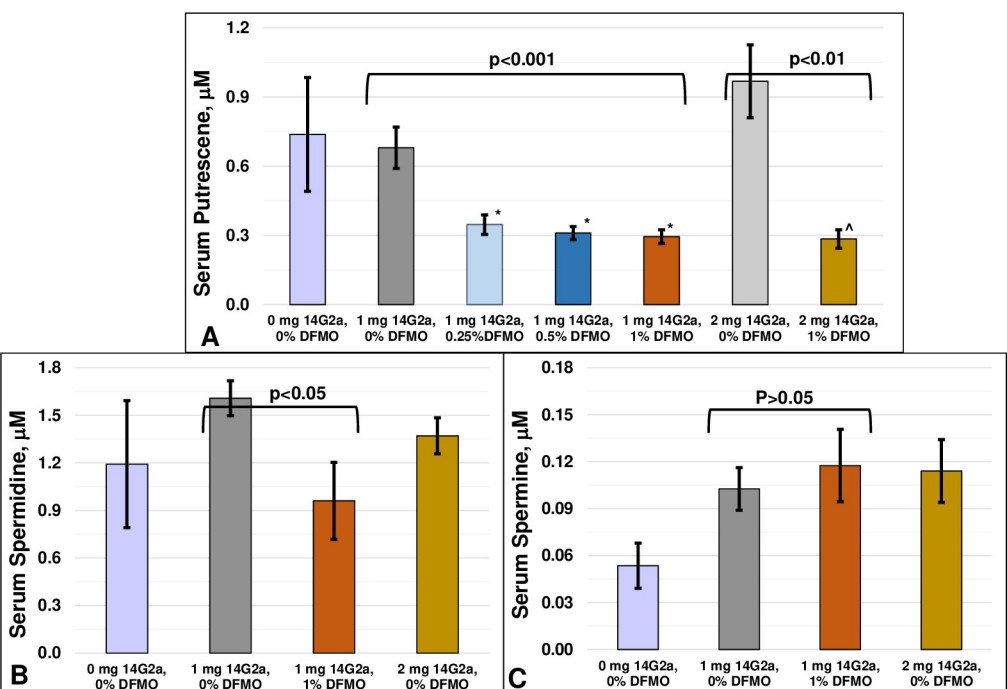

**Fig 3. Polyamine levels in rat blood after anti-GD2 injections.** Polyamine levels were measured in serum from rats injected with 0, 1 or 2 mg/kg 14G2a and allowed to drink water with various concentrations of DFMO and are presented as mean ± SEM. **A)** Putrescine levels were reduced by DFMO in animals injected with 1 mg/kg (p<0.001, one-way ANOVA). *p<0.01, each vs. 0% DFMO, Fishers LSD post-hoc analysis; ^p>0.05, 1 mg/kg/1% DFMO vs. 2 mg/kg/1% DFMO, t-test; **B)** Spermidine levels were significantly reduced by 1% DFMO in the animals injected with 1 mg/kg 14G2a (p<0.05, t-test); **C)** Spermine levels were not affected by DFMO treatment (p>0.05, t-test).

thus it would be expected to be most affected by ODC inhibition. Spermidine is synthesized next in the polyamine cascade and was also significantly lower in the 1 mg/kg 14G2a, 1% DFMO-treated animals (Fig 3B, mean ± SE, 1.61 μM ± 0.11 (0%, N = 14) and 0.96 ± 0.24 μM (1%, N = 8); p<0.05, t-test). In contrast, no changes in spermine were observed between animals injected with 1 mg/kg and drinking water without or with 1% DFMO (0.10 ± 0.01 μM (0%, N = 14) vs 0.12 ± 0.02 μM (1%, N = 8) (p>0.05, Fig 3C).

We next compared putrescine levels in control and 1% DFMO animals given the higher dosage of 14G2a. Pretreatment with 1% DFMO not only significantly reduced mean putrescine levels in the 2 mg/kg treated animals (0.97+/- 0.16 (0%, N = 6) vs 0.28+/- 0.04 (1%, N = 12), p<0.01. t-test), but reduced the mean to levels indistinguishable from animals injected with 1 mg/kg 14G2a/1% DFMO (p>0.05, t-test) (Fig 3A). This suggests that 1% DFMO achieved maximal reduction of putrescine at both 14G2a dosages.

## Discussion

Anti-GD2 therapy with dinutuximab has been validated as efficacious in the treatment of children with high risk neuroblastoma in remission and with relapsed and refractory disease, although there remain a number of patients who do not respond and/or relapse [4, 5, 7, 8]. Additional therapeutic approaches and/or combination therapies are clearly needed to further improve outcome. However, dose-escalation of dinutuximab is challenging due to the severe pain caused by the anti-GD2. Patients suffer from severe visceral pain and whole-body allodynia that begins within an hour after initiation of dinutuximab infusion [26].

In the rat, GD2 ganglioside is located on Schwann cells located along the peripheral nerve fibers [27]. Systemic administration of either a murine (14G2a) or a murine/human chimeric (ch14.18, (dinutuximab)) antibody elicits reduced mechanical withdrawal threshold, i.e., allodynia, and whole-body touch evoked agitation [25, 28]. Coincident with antibody-induced pain behavior in the rat model, distal C fibers that are surgically disconnected from their cell bodies and the central nervous system, but are still attached to the skin, develop high frequency spontaneous activity after anti-GD2 injection [29]. This may be due to either an excitatory effect of the 14G2a on the peripheral nerve fiber or to the cutaneous nerve endings (Fig 4). As the pain behavior is blocked by systemic pre-treatment with a complement C5 antagonist [30], the assumption is that the nociceptive actions produced by anti-GD2 therapy occurs downstream of an antibody/antigen interaction on the non-myelinating Schwann cells that surround the C fibers. The situation is more complicated clinically, where malignancies enriched with the GD2 ganglioside, e.g. neuroblastoma, provide more substrate for this reaction. The preponderant localization of neuroblastoma tumors in proximity to visceral C fibers may be the proximate cause of the particularly intense visceral pain seen in these patients.

Potential mechanisms by which pain behavior may be modulated by 14G2a and DFMO. Polyamine (PA) injection into the skin [16] or around the spinal cord [31] both induce pain behavior while DFMO decreases pain behavior [16]. Systemic administration of anti-GD2 induces activation of C fibers and also causes an increase in pain behavior [25, 28, 29]. Nerve injury increases ODC and PA at the dorsal root ganglia (DRG) [32]. In the skin, pain behavior is blocked by pretreatment with antagonists to TRPV1R [33], which are found on a subpopulation of peptide (substance P) and calcitonin gene-related protein (CGRP)) expressing C fibers. Importantly, functional elimination of these fibers also blocks anti-GD2-induced pain behavior [34]. *Spinal* administration of antagonists to the NMDAR, a glutamate receptor subtype, were highly effective in blocking spinal polyamine induced pain behavior [31], though the *peripheral* co-administration of NMDAR antagonists with polyamines had no effect on the evoked pain behavior [33]. Green lines are excitatory, red lines are inhibitory.

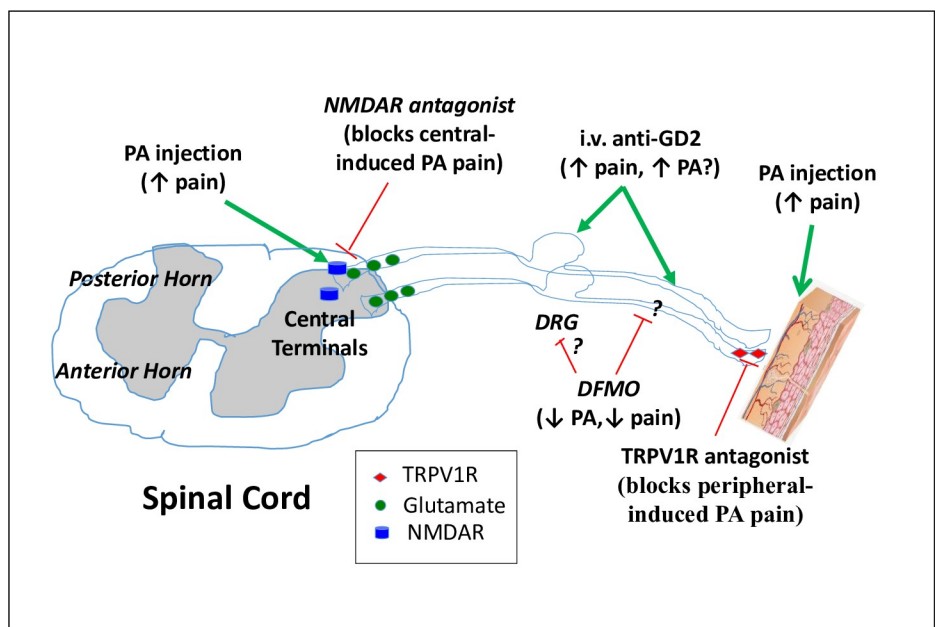

**Fig 4. Model of pain pathways from anti-GD2 and polyamines.**

Our data confirm that anti-GD2 is evoking a pain response in the absence of tumors, which is consistent with preclinical studies from our group [25] and the pain observed in neuroblastoma patients without detectable disease who were receiving maintenance therapy of anti-GD2 [4]. We hypothesized that even in the absence of tumor cells, treatment with antibodies against GD2 may be directly or more probably indirectly activating the ornithine pathway and newly synthesized polyamines and/or released intracellular polyamines may be contributing to the observed pain. This assumption was supported by the work of Mäntyselkä et al [35], who showed that ornithine levels in adults with persistent musculoskeletal pain were elevated compared to patients with non-persistent pain, and both were higher than those seen in a control population. Other studies have also demonstrated elevated plasma ornithine in patients with complex regional pain syndrome compared to that of ornithine levels in controls [36]. Downstream polyamine involvement in several pro-nociceptive receptors/systems is also well substantiated. Spermine and spermidine induce a biphasic dose-dependent activation/inactivation of NMDA receptors (NMDAR) in spinal cord and cerebral cortex [37]. Similarly, spinal spermine induced biphasic effects on pain behavior with pronounced hyperalgesia at higher doses [31]. Spinal polyamine elicited pain behavior was blocked by a variety of NMDA receptor antagonists, including notably ifenprodil, an antagonist of the GluN2B subunit [31, 38]. It is unknown whether these spinal actions are due to presynaptic or postsynaptic actions. However, since the antibody does not cross the blood brain barrier in an animal with no inflammation or malignancies, the initial step in antibody-induced allodynia in our study is likely due to an action at or in the immediate vicinity of the peripheral nerve.

The novel findings from this study are that 1% DFMO treatment significantly reduced anti-GD2-induced pain behavior and that this reduction was seen in the face of antibody dose escalation, thus our original hypothesis was confirmed. Results concerning our second contention that there was an association between magnitude of pain behavior and plasma levels of polyamines are less clear. Blood sampling was done at the end of the study, 24 hours after the GD2 injection, to minimize confounding effects on animal behavior. As animals remained on water

containing DFMO during the entire time, polyamine levels would remain suppressed even as allodynia waned. However, it is unlikely that this confounded their correlation with peak allodynia as the animals had already been on DFMO for 7–10 days. Differences between the peak reduction in pain and PA levels might reflect different dose response curves for actions of DFMO on pain versus polyamine reduction. It is notable that in two independent studies of pain and polyamine levels, no direct correlation between expressed pain magnitude and measured ornithine was observed [16, 36]. Our study also does not exclude the possibility of discrete local changes in polyamine levels after anti-GD2 injection, especially considering the limited expression of the GD2 antigen outside of nerve fibers in the absence of cancer, and the fact that nerve injury has been shown to be associated with an increase in ODC and polyamines at the dorsal root ganglia [32, 39]. What we can conclude from our study is that DFMO treatment in rats given anti-GD2 produced a clear decrease in pain and polyamines.

Actions of DFMO on pathways besides or in addition to polyamines must also be considered. Caspase-12 is considered to be pro-inflammatory and increases within the nociceptive matrix after nerve injury [40]. Agents such as cannabinoids and ozone which reduce ER stress reduce both pain behavior and the increase in caspase-12 [41, 42]. This is significant as DFMO has also been reported to reduce the expression of caspase-12 in some animal models involving endoplasmic reticulum stress mediation pathways [43]. In a study of the effects of DFMO on cardiac hypertrophy in rats, DFMO not only inhibited ODC and decreased polyamine levels, but also decreased the expression of GRP78, protein kinase R (PKR)-like endoplasmic reticulum kinase (PERK), calreticulin, and caspase-12 [43]. However, it is unknown if these are direct actions of DFMO or secondary actions resulting from ODC inhibition and/or decreases in polyamine levels. These studies demonstrate that the effects of DFMO on pain may go beyond ODC inhibition, and offer the possibility that the depletion of polyamines may not be the (sole) mechanism by which DFMO reduces anti-GD2 induced allodynia.

A number of clinical studies are investigating DFMO as a therapeutic adjunct to chemotherapy for patients with a number of different cancers (ClinicalTrials.gov). In particular, DFMO has already shown promise in the treatment of high-risk neuroblastoma when given during the *post*-maintenance phase when compared to historic controls [44]. However, this is the first study to demonstrate that DFMO may also be associated with the amelioration of the clinically observed pain induced by anti-GD2 therapy. The biologically effective anti-cancer blood levels of DFMO are in the 50–150 μM range for both adult and pediatric cancers ([15] and references therein). These concentrations are slightly higher than we have observed for rats on 1% DFMO, the group that showed the greatest anti-allodynic benefit. This suggests that DFMO concentrations with anti-tumor activity should be sufficient to also confer anti-allodynic benefits and are not only readily attainable, but may allow for an increase in anti-GD2 dosage as well.

## Conclusion

Currently, neuroblastoma is the only disease approved for anti-GD2 therapy. However, dinutuximab is also being investigated for use in other cancers that express GD2, such as small cell lung cancer and osteosarcoma [45]. DFMO is showing promise as an anti-cancer adjunct when used in combination with various chemotherapeutics and during the post-maintenance phase of ant-GD2 therapy of neuroblastoma. Thus, there may be multiple benefits of DFMO in combination with anti-GD2: 1) reduced pain allows for increased dosage and possibly increased anti-GD2 efficacy; 2) reduced pain may expand the patient population and cancer types amiable for anti-GD2 therapy; and 3) DFMO may contribute an anti-cancer benefit in addition to or independent of its anti-allodynic effect. In summary, using DFMO as an adjunct

to dinutuximab may allow dose escalation for neuroblastoma patients on anti-GD2 immuno-therapy and a sufficient reduction in pain such that new patient populations may utilize this therapy given the more acceptable side effect profile.

## Supporting information

**S1 Table. Scheduled acquisition times of putrescine and DFMO.** Ion (m/z) transitions monitored, scheduled acquisition times (minutes), analyzer parameters (declustering (DP), entrance (EP) and collision cell (CXP) potentials) and analytical range (micromoles per liter; μL) for putrescine, difluoromethylornithine (DFMO), and stable-isotope labeled internal standards, 1,1,2,2,3,3,4,4-$^2$H$_8$-putrescine (d8-putrescine) and 5,5-$^2$H$_2$-ornitine (d2-ornithine). Numerals -1 indicate the transition used for quantification, and -2 the one used for confirmation. (DOCX)

**S1 Method. Putrescene and DFMO measurement by HPLC and LC-MS/MS.** (DOCX)

**S2 Method. Putrescene, spermidine and spermine measurement by HPLC and LC-MS/MS using derivatives.** (DOCX)

**S1 Appendix. The ARRIVE guidelines checklist.** (PDF)

**S1 Raw data. Rat serum polyamine and DFMO levels.** (XLSX)

## Author Contributions

**Conceptualization:** Mitchell B. Diccianni, Alice L. Yu, Linda S. Sorkin.

**Data curation:** Mitchell B. Diccianni, Linda S. Sorkin.

**Formal analysis:** Mitchell B. Diccianni, Linda S. Sorkin.

**Funding acquisition:** Mitchell B. Diccianni, Alice L. Yu.

**Investigation:** Mitchell B. Diccianni, Katarzyna Kempińska, Jon A. Gangoti, Linda S. Sorkin.

**Methodology:** Mitchell B. Diccianni, Jon A. Gangoti, Linda S. Sorkin.

**Project administration:** Mitchell B. Diccianni, Linda S. Sorkin.

**Supervision:** Mitchell B. Diccianni, Linda S. Sorkin.

**Validation:** Mitchell B. Diccianni, Linda S. Sorkin.

**Writing – original draft:** Mitchell B. Diccianni, Jon A. Gangoti, Linda S. Sorkin.

**Writing – review & editing:** Mitchell B. Diccianni, Alice L. Yu, Linda S. Sorkin.

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
