## [Decision Letter · Decision Letter 0]

17 Mar 2020

PONE-D-20-03409

Anti-GD2 induced allodynia in rats can be reduced by pretreatment with DFMO

PLOS ONE

Dear Dr. Diccianni,

Thank you for submitting your manuscript to PLOS ONE. After careful consideration, we feel that it has merit but does not fully meet PLOS ONE’s publication criteria as it currently stands. Therefore, we invite you to submit a revised version of the manuscript that addresses the points raised during the review process.

We would appreciate receiving your revised manuscript by May 01 2020 11:59PM. To enhance the reproducibility of your results, we recommend that if applicable you deposit your laboratory protocols in protocols.io, where a protocol can be assigned its own identifier (DOI) such that it can be cited independently in the future. For instructions see: http://journals.plos.org/plosone/s/submission-guidelines#loc-laboratory-protocols

We look forward to receiving your revised manuscript.

Kind regards,

Giuseppe Biagini, MD

Academic Editor

PLOS ONE

Journal Requirements:

2. At this time, we request that you  please report additional details in your Methods section regarding animal care, as per our editorial guidelines. Specifically, please state the number of mice used in the study. Thank you for your attention to these requests.

3. Please note that PLOS does not permit references to “data not shown.” or "unpublished data". Authors should provide the relevant data within the manuscript, the Supporting Information files, or in a public repository. If the data are not a core part of the research study being presented, we ask that authors remove any references to these data.

4. At this time, we ask that you please the product number of the DFMO obtained from Professor Patrick M. Woster in your study.

Reviewers' comments:

Reviewer's Responses to Questions

**Comments to the Author**

1. Is the manuscript technically sound, and do the data support the conclusions?

Reviewer #1: Partly

Reviewer #2: Yes

2. Has the statistical analysis been performed appropriately and rigorously? 

Reviewer #1: Yes

Reviewer #2: Yes

3. Have the authors made all data underlying the findings in their manuscript fully available?

Reviewer #1: Yes

Reviewer #2: Yes

4. Is the manuscript presented in an intelligible fashion and written in standard English?

Reviewer #1: Yes

Reviewer #2: No

5. Review Comments to the Author

Reviewer #1: The ms of Diccianni et al is directed to test the efficacy of DFMO against the allodynia induced by anti-GD2 treatment. Behavioral tests show antiallodynic effects of DFMO. Furthermore, a concomitant reduction in plasmatic levels of putrescene and spermidine was measured using high performance liquid chromatography and mass spectrometry. The objectives of the work are clear and straightforward, and the experiments seem adequately performed. The results are in line to previous reports that have shown analgesic effects of DFMO in models of pain, making them extensible to this model of anti-GD2 allodynia.

In my opinion the work is correctly made, and my unique concern about the ms is related to the insistence of the authors to justify a causal relation between antiGD2, polyamines and pain, that is not directly supported by their data. As an example “treatment with antibodies against GD2 are activating the ornithine pathway and/or releasing intracellular polyamines”. Their arguments to draw this relation are reasonably explained in basis of existing literature, but they are speculative considering the data showed, and hence the authors should be more cautious and consider alternative explanations.

AntiGD2 do not increase polyamines, at least at 24h. The argument that poliamines may be elevated at early steps is reasonable, but need to be demonstrated. So the authors may consider to make an analysis of polyamines at earlier time points, this will strength their arguments. Since only 100 microlitres of plasma seem necessary for the determination, and the concentration of polyamines varies considerably between animals, if possible, the most interesting approach could be to compare the values before and after antiGD2 in the same animals for one of the polyamines. To stablish a causal relation, polyamines should be elevated from the time point that allodynia can be detected (or even before).

Alternatively, if the authors are not able to make this assay, they should give more weight to other putative explanations that are only briefly mentioned in the current version. Although it seems less likely, anti-GD2 may induce allodynia by a mechanism independent of polyamines, and DFMO may also reduce allodynia by mechanisms independent of polyamines reduction (despite being able to reduce them).

Other comments:

If possible, it would be of interest that the authors may explain how they believe that results using an acute dose of anti-GD2 can be extrapolated to long lasting treatment. Would they expect dissimilarities?

In the last sentence of the results section the authors claim that since the reduction in putrescine levels produced by 1% DFMO was similar between animals treated with 1 and 2 mg/kg of anti-GD2 “this suggests that 1% DFMO has achieved maximal reduction of putrescine at both 14G2a dosages”. Since the levels of putrescine shown are not different between both dosages of anti-GD2, it is reasonable that the same concentration of DMFO had the same effect as well, and, from my point of view, it does not exclude that higher concentrations of DMFO may produce higher reductions. In addition, the variability between subjects is perhaps very strong to make this type of assumption.

ODC abbreviation need to be explained in the abstract since sometimes is read alone.

Take into account that the resolution of the figures is rather bad, even downloading them. For figures 2 A and C, perhaps may be valuable to indicate the statistical significance at the different time points, it would be more descriptive. In addition, an asterisk seems to be absent in fig 2B (“there was a significant difference only between the control (non-DFMO) group and the two DFMO groups (both p<0.05)”).

Reviewer #2: The manuscript by Diccianni and collaborators entitled Anti-GD2 induced allodynia in rats can be reduced by pretreatment with DFMO reported data regarding the effect of DFMO on allodynia induced by Anti-GD2 therapy

Manuscript is quite confusing.

The authors are encouraged to more carefully develop their results along the following lines:

Authors need to choose a single dose of 14G2a to show. Why they reported both 1mg and 2 mg?

They already reported that intravenous injection of 1 mg/kg 14G2a, a

dosage within the range used in children, into rat tail vein induces withdrawal

responses at pressures that are normally innocuous, indicative of pain behavior

(allodynia) (ref 23).

0.25 and 0.5 of DFMO are reported together. It is not right to explore a dose related response. Please carefully revise.

Figure 2 need to be before figure 1

Please carefully revise all the ms to point on the major results.

Please also revise the discussion by reducing and focusing on the advance reported by the proposed study

6. PLOS authors have the option to publish the peer review history of their article (what does this mean?). If published, this will include your full peer review and any attached files.

Reviewer #1: No

Reviewer #2: No

---

## [Author Response · Author response to Decision Letter 0]

20 Apr 2020

Dear Editor and esteemed reviewers, 

We thank you all for your thorough review and critique of our manuscript entitled: “Anti-GD2 induced allodynia can be reduced in rats by pretreatment with DFMO” by Diccianni et al. for consideration of publication in PLoS1 as a Research Article. We have addressed the concerns of the reviewer and present this point by point response in red in this cover letter. We have modified the results section as recommended, and have extensively rewritten the discussion. We have uploaded both a tracked and tracking accepted version of the manuscript. We hope you find this revised version acceptable for publication. 

Sincerely yours,

Mitchell B. Diccianni

 

Journal Requirements:

These links do not work. However, we have followed the correct format. 

2. At this time, we request that you please report additional details in your Methods section regarding animal care, as per our editorial guidelines. Specifically, please state the number of mice used in the study. Thank you for your attention to these requests.

The ARRIVE guidelines have been followed and an ARRIVE checklist is included in the supplemental materials. This fact is now mentioned in the Methods section (page 6) and included as S1 appendix.

We have now added a total to the methods section (page 6), while the individual breakdown of animals per treatment group remains throughout the results section.

3. Please note that PLOS does not permit references to “data not shown.” or "unpublished data". Authors should provide the relevant data within the manuscript, the Supporting Information files, or in a public repository. If the data are not a core part of the research study being presented, we ask that authors remove any references to these data.

 These have all been removed or referenced in the revised version. 

4. At this time, we ask that you please the product number of the DFMO obtained from Professor Patrick M. Woster in your study.

 Page 8: There is no product number. We have clarified with Dr. Woster and modified the phrase as follows: “DFMO was synthesized by Genzyme Corporation (Lot #PD39CF) and graciously provided to us by Dr. Patrick Woster of the Medical University of South Carolina, Charleston, South Carolina”.

 See #3 above.

 Added.

Comments to the Author

Reviewer#1:

The ms of Diccianni et al is directed to test the efficacy of DFMO against the allodynia induced by anti-GD2 treatment. Behavioral tests show antiallodynic effects of DFMO. Furthermore, a concomitant reduction in plasmatic levels of putrescene and spermidine was measured using high performance liquid chromatography and mass spectrometry. The objectives of the work are clear and straightforward, and the experiments seem adequately performed. The results are in line to previous reports that have shown analgesic effects of DFMO in models of pain, making them extensible to this model of anti-GD2 allodynia.

In my opinion the work is correctly made, and my unique concern about the ms is related to the insistence of the authors to justify a causal relation between antiGD2, polyamines and pain, that is not directly supported by their data. As an example “treatment with antibodies against GD2 are activating the ornithine pathway and/or releasing intracellular polyamines”. Their arguments to draw this relation are reasonably explained in basis of existing literature, but they are speculative considering the data showed, and hence the authors should be more cautious and consider alternative explanations.

The reviewer is correct in that our data document that pain is prevented and PA levels reduced, but the relationship between the two actions is less than clear. Inhibition of ODC and the subsequent reduction in polyamine levels, in particular putrescene, is the best characterized role for DFMO. Other actions of DFMO are either less characterized or may be secondary actions of ODC inhibition and polyamine depletion. However, both reviewers felt it was necessary to expand these points, which we do on pages 19-21. 

We modified the sentence on page 19 “…treatment with antibodies against GD2 are activating the ornithine pathway…” to the more speculative “We hypothesized that treatment with antibodies against GD2 may be activating the ornithine pathway” and rewrote the paragraph surrounding.

AntiGD2 do not increase polyamines, at least at 24h. The argument that poliamines may be elevated at early steps is reasonable, but need to be demonstrated. So the authors may consider to make an analysis of polyamines at earlier time points, this will strength their arguments. Since only 100 microlitres of plasma seem necessary for the determination, and the concentration of polyamines varies considerably between animals, if possible, the most interesting approach could be to compare the values before and after antiGD2 in the same animals for one of the polyamines. To stablish a causal relation, polyamines should be elevated from the time point that allodynia can be detected (or even before).

Alternatively, if the authors are not able to make this assay, they should give more weight to other putative explanations that are only briefly mentioned in the current version. Although it seems less likely, anti-GD2 may induce allodynia by a mechanism independent of polyamines, and DFMO may also reduce allodynia by mechanisms independent of polyamines reduction (despite being able to reduce them).

The reviewer is also correct in that at least at 24h, our data fail to show an increase in polyamine levels after anti-GD2 injections. Obtaining blood at peak allodynia would have required either a chronically placed catheter to withdraw blood, or infraorbital blood sampling. The first method would have required a larger sample to be withdrawn to account for the dead space in the catheter and would require the animals to be housed singly (they chew on each other’s catheters), the second method is likely to have influenced behavioral testing and would likely have caused problems with the animal care committee, which is reluctant to approval for additional procedures in studies with maintained allodynia/pain. Since our primary clinically-motivated objective was to determine whether DFMO would reduce anti-GD2 allodynia, we did not perform mid-experiment blood sampling as we did not want to risk jeopardizing these behavioral results.

That said, we are not sure the results would have been any different. The animals were on DFMO containing water for 7-10 days. We expect polyamines would have been suppressed and have remained suppressed prior to and during the study. This is discussed on pages 20-21

In the original version, we discussed “alternative” mechanisms only briefly. For this revision, we have expand the discussion of “other putative explanations” (i.e. speculate more, page 20-21) while balancing Reviewer’s 2 concern to focus the discussion.

We have edited “…this is the first study to demonstrate that depletion of polyamines…” to “…this is the first study to demonstrate that DFMO …”) on page 22.

Other comments:

If possible, it would be of interest that the authors may explain how they believe that results using an acute dose of anti-GD2 can be extrapolated to long lasting treatment. Would they expect dissimilarities?

Another excellent point! Our introduction (page 3) explains the limits of anti-GD2 therapy. We have added a clinical study of DFMO in neuroblastoma to the discussion (pg 22) and a summary in the conclusion as to why the two could be better together than alone (pg 22-23).

In the last sentence of the results section the authors claim that since the reduction in putrescine levels produced by 1% DFMO was similar between animals treated with 1 and 2 mg/kg of anti-GD2 “this suggests that 1% DFMO has achieved maximal reduction of putrescine at both 14G2a dosages”. Since the levels of putrescine shown are not different between both dosages of anti-GD2, it is reasonable that the same concentration of DMFO had the same effect as well, and, from my point of view, it does not exclude that higher concentrations of DMFO may produce higher reductions. In addition, the variability between subjects is perhaps very strong to make this type of assumption.

True. Putrescene levels seem to be maximally decreased at the lowest [DFMO] used while the relief of allodynia show a relationship with [DFMO] used (Figure 1A). Thus a higher concentration of DFMO may suppress allodynia further, something particularly beneficial if a higher dose of anti-GD2 is to be used. However, DFMO is bitter. In human studies, the DFMO is given mixed with juice to increase palatability. Our study design required the passive consumption of DFMO-containing water. The increased bitterness of a 2% DFMO solution may result in the animals drinking less. In this case, the effects of [DFMO] at 2% may be only slightly or not at all different than at 1%. 

ODC abbreviation need to be explained in the abstract since sometimes is read alone.

 OK, this has been added.

Take into account that the resolution of the figures is rather bad, even downloading them.

 OK-we used PACE this time around.

For figures 2 A and C, perhaps may be valuable to indicate the statistical significance at the different time points, it would be more descriptive. In addition, an asterisk seems to be absent in fig 2B (“there was a significant difference only between the control (non-DFMO) group and the two DFMO groups (both p<0.05)”).

We did not include asterisks for all the individual points as it made the figure very messy. The text states (pg 12) and a new S1 Table shows where significance lies. 

A single asterisk in Fig 2B (now Figure 1B per Reviewer 2’s critique) is correct.

Reviewer #2: The manuscript by Diccianni and collaborators entitled Anti-GD2 induced allodynia in rats can be reduced by pretreatment with DFMO reported data regarding the effect of DFMO on allodynia induced by Anti-GD2 therapy

Manuscript is quite confusing.

We are sorry the reviewer found the manuscript quite confusing. We tried to present a manuscript that was concise and straight forward: Oral DFMO reduces anti-GD2 induced allodynia (novel finding); DFMO reduces polyamine levels (as expected); the reduction in polyamines levels did not directly correlate with the reduction of allodynia (novel finding requiring some speculation). In this revision, we have tried to clarify these thoughts further.

The authors are encouraged to more carefully develop their results along the following lines:

Authors need to choose a single dose of 14G2a to show. Why they reported both 1mg and 2 mg? They already reported that intravenous injection of 1 mg/kg 14G2a, a dosage within the range used in children, into rat tail vein induces withdrawal responses at pressures that are normally innocuous, indicative of pain behavior (allodynia) (ref 23).

We strongly disagree with this thought. Dinutuximab is an excellent but imperfect therapeutic for high risk neuroblastoma treatment. However, late failures occur. Higher dosages cannot be given due to the toxicity profile. In particular, whole body, opiate resistant allodynia/pain prevents an increase in dinutuximab dosage. We show in this manuscript that not only does an increase in anti-GD2 cause more allodynia, but this enhanced allodynia can be attenuated by DFMO. When given in combination, DFMO has the potential to allow anti-GD2 to be used at increased dosages. This is a major conclusion.

0.25 and 0.5 of DFMO are reported together. It is not right to explore a dose related response. Please carefully revise. Figure 2 need to be before figure 1

We have ungrouped the data and swapped the order of the first two figures.

Please carefully revise all the ms to point on the major results.

Please also revise the discussion by reducing and focusing on the advance reported by the proposed study.

We have extensively revised the discussion to focus more but, in deference to reviewer #1, have added role for DFMO and anti-GD2 therapy (page 22) and speculate on alternative mechanisms of action (page 21).

---

## [Decision Letter · Decision Letter 1]

12 May 2020

PONE-D-20-03409R1

Anti-GD2 induced allodynia in rats can be reduced by pretreatment with DFMO

PLOS ONE

Dear Dr- Diccianni,

Thank you for submitting your manuscript to PLOS ONE. After careful consideration, we feel that it has merit but does not fully meet PLOS ONE’s publication criteria as it currently stands. Therefore, we invite you to submit a revised version of the manuscript that addresses the points raised during the review process.

We would appreciate receiving your revised manuscript by Jun 26 2020 11:59PM. To enhance the reproducibility of your results, we recommend that if applicable you deposit your laboratory protocols in protocols.io, where a protocol can be assigned its own identifier (DOI) such that it can be cited independently in the future. For instructions see: http://journals.plos.org/plosone/s/submission-guidelines#loc-laboratory-protocols

We look forward to receiving your revised manuscript.

Kind regards,

Giuseppe Biagini, MD

Academic Editor

PLOS ONE

Reviewers' comments:

Reviewer's Responses to Questions

**Comments to the Author**

1. If the authors have adequately addressed your comments raised in a previous round of review and you feel that this manuscript is now acceptable for publication, you may indicate that here to bypass the “Comments to the Author” section, enter your conflict of interest statement in the “Confidential to Editor” section, and submit your "Accept" recommendation.

Reviewer #1: (No Response)

Reviewer #2: All comments have been addressed

2. Is the manuscript technically sound, and do the data support the conclusions?

Reviewer #1: Partly

Reviewer #2: Partly

3. Has the statistical analysis been performed appropriately and rigorously? 

Reviewer #1: I Don't Know

Reviewer #2: Yes

4. Have the authors made all data underlying the findings in their manuscript fully available?

Reviewer #1: Yes

Reviewer #2: Yes

5. Is the manuscript presented in an intelligible fashion and written in standard English?

Reviewer #1: Yes

Reviewer #2: Yes

6. Review Comments to the Author

Reviewer #1: The revised version of the ms of Diccianni et al entitled “Anti-GD2 induced allodynia in rats can be reduced by pretreatment with DFMO” tested the validity of DFMO against the allodynia induced by anti-GD2, a pharmacological treatment for neuroblastoma. They show that animals allowed to drink water with DFMO showed reduced nociceptive behaviors after anti-GD2 treatment and reduced levels of polyamines at 24 hours. Previous reports have shown analgesic effects of DFMO as well as the relation between polyamines and pain. The use of DFMO can provide a benefit for patients treated with anti-GD2.

I have some concerns about statistical analysis. Now seems clear that comparisons between three or more groups were always done using One-way ANOVAs (please include “One-way” in methods). The authors may consider to include the degrees of freedom and the statistic’s values, in addition to the p value. Some of the interpretations made are perhaps not adequate considering the analysis done and other test (perhaps Two-way ANOVA) that can compare different treatments should be used to support the description of some results:

- Page 11/31 “At every time point measured from 1h until the end of the study, animals given 2 mg/kg had lower average withdrawal thresholds (more allodynia) than those given the lower dose” and “These data demonstrate a dose dependent increase in pain behavior elicited by 14G2a, whereas an increased dosage of 14G2a increased the magnitude … of the pain behavior …” what test was used to support these commentaries.

- Page12/31 “exhibit significant pain behavior after injection with 1 mg/kg 14G2a …, though sensitivity was reduced … relative to non-DFMO watered animals” Please include the test used to affirm that sensitivity was reduced.

- Figure1, legend page 11/31, “both conditions versus pre-treatment” and page 12-13/31 “withdrawal thresholds were not significantly different from pre-injection levels at any timepoint (p>0.05, repeated measures ANOVA” ANOVA compares the means of ALL groups, no specifically versus pre-treatment.

- Page 13/31 “DFMO again significantly attenuated allodynia (p<0.001, one-way repeated measures ANOVA” If the authors want to compare the results with or without DFMO (in order to say that attenuates allodynia) they should other type of analysis comparing both treatments (14G2a with and without DFMO).

- Other questions related to statistic: Figure1, legend page 11/31, “1 mg/kg injection of 14G2a (both p<0.001,” what “both” means and “*p<0.05 versus 0% DFMO only” what versus 0%?. Page 12/31 “N=14, p<0.001,” why including N=14? What does p<0.001 stand for?

The objective two “determine if there is an association between plasma levels of polyamines and magnitude of pain behavior” is problematic. Results from animals treated with 14G2a (but not with DFMO) and controls showed no change at 24h and no earlier time points were analyzed as a proof of concept for this relation. In addition, the authors now indicate that a local change might take place (even during DFMO treatment) suggesting that plasma measurement could be an incorrect way to stablish a causal relation between polyamines and anti GD2 induced pain. These two arguments make difficult to support the speculation of such association, and the inclusion of both effects together, for example in the abstract “Administration of DFMO attenuated the enhanced sensitivity. Consistent with the known actions of DFMO on ornithine decarboxylase (ODC), serum putrescene and spermidine levels were significantly reduced by DFMO”, may give an incorrect idea of what the work does really demonstrate and confound the reader.

In the new version, the results are even more complicated including statistics at different time points and concentrations, and some of them seem repeated in figure legends, including statistics. The authors may explore other forms of writings the results in a more clear and direct way and take advantage of visual support (figures-tables) to discharge this section. Consider to include it inside the manuscript, better than in supplementary material, since it is something relevant and helpful for the reader.

After rewriting the discussion, these sentences “Spermine and spermidine induce a biphasic dose dependent activation/inactivation of NMDA receptors (NMDAR) on cortical neurons and induce pain behavior that can be blocked with NMDAR antagonists (33, 34). It is unknown whether these spinal actions are due to presynaptic or postsynaptic actions.” may not be correct. Neither of these works seem to look at the spinal cord. Ref 34 is a study at a peripheral level, and, if fact, shows that NMDAR are not implicated in the nociceptive actions of polyamines (as the authors indicate in fig 4). Perhaps the work by Kolhekar et al. 1994 (Neuroscience 63(4):925-36) could be more adequate to support their affirmation. But anyway the authors must be attentive to this.

Minors:

In page 7/31 is written isoflorane instead of isoflurane

In page 10/31 is written Prizm instead of Prism

Reviewer #2: Authors reply to the observations. please spell out all the abbreviations the first time they appear

Ex in the abstract DFMO is spelled out the second time please correct.

7. PLOS authors have the option to publish the peer review history of their article (what does this mean?). If published, this will include your full peer review and any attached files.

Reviewer #1: No

Reviewer #2: No

---

## [Author Response · Author response to Decision Letter 1]

10 Jun 2020

Review Comments to the Author

 Reviewer #1: The revised version of the ms of Diccianni et al entitled “Anti-GD2 induced allodynia in rats can be reduced by pretreatment with DFMO” tested the validity of DFMO against the allodynia induced by anti-GD2, a pharmacological treatment for neuroblastoma. They show that animals allowed to drink water with DFMO showed reduced nociceptive behaviors after anti-GD2 treatment and reduced levels of polyamines at 24 hours. Previous reports have shown analgesic effects of DFMO as well as the relation between polyamines and pain. The use of DFMO can provide a benefit for patients treated with anti-GD2.

• I have some concerns about statistical analysis. Now seems clear that comparisons between three or more groups were always done using One-way ANOVAs (please include “One-way” in methods). The authors may consider to include the degrees of freedom and the statistic’s values, in addition to the p value. Some of the interpretations made are perhaps not adequate considering the analysis done and other test (perhaps Two-way ANOVA) that can compare different treatments should be used to support the description of some results:

“One-way” has been added to our ANOVAs. 

Regarding the time compressed AUC behavior data, the use of AUC is quite common for this type of data and allows us to perform within and between comparisons similar to that of the two-way ANOVA. This analysis is forgiving of individual variations in the behavioral time courses among animals as well as allowing for differences in Ns.

• Page 11/31 “At every time point measured from 1h until the end of the study, animals given 2 mg/kg had lower average withdrawal thresholds (more allodynia) than those given the lower dose” and “These data demonstrate a dose dependent increase in pain behavior elicited by 14G2a, whereas an increased dosage of 14G2a increased the magnitude … of the pain behavior …” what test was used to support these commentaries.

This qualitative observation is supported by a comparison of the AUCs at different doses of 14G2a without DFMO compared by t-tests (page 11).

• Page12/31 “exhibit significant pain behavior after injection with 1 mg/kg 14G2a …, though sensitivity was reduced … relative to non-DFMO watered animals” Please include the test used to affirm that sensitivity was reduced.

This is part of the repeated measures ANOVA (clarified in the sentence). We now also reference Table 1 (formerly S1 Table) for added clarification. 

• Figure1, legend page 11/31, “both conditions versus pre-treatment” and page 12-13/31 “withdrawal thresholds were not significantly different from pre-injection levels at any timepoint (p>0.05, repeated measures ANOVA” ANOVA compares the means of ALL groups, no specifically versus pre-treatment.

Legend page 11/31: Both conditions refers to the fact that both 0%/2mg and 1%/2 mg were significantly different than pretreatment by repeated measures ANOVA, as stated.

page 12-13/31: The second point of concern deals solely with 1 mg/kg and 1% DFMO. Under these conditions, a repeated measures ANOVA analysis was non-significant, demonstrating that withdrawal sensitivity was completely attenuated at all time points. This is also exactly what is stated and the point we wanted to make. 

• Page 13/31 “DFMO again significantly attenuated allodynia (p<0.001, one-way repeated measures ANOVA” If the authors want to compare the results with or without DFMO (in order to say that attenuates allodynia) they should other type of analysis comparing both treatments (14G2a with and without DFMO).

We have clarified this sentence. 

• Other questions related to statistic: Figure1, legend page 11/31, “1 mg/kg injection of 14G2a (both p<0.001,” what “both” means and “*p<0.05 versus 0% DFMO only” what versus 0%?. Page 12/31 “N=14, p<0.001,” why including N=14? What does p<0.001 stand for?

We removed the word “both”. 

*p<0.05 versus 0%, the asterisk refers to 0% vs. 1% in Figure 1B. We have added text to additionally clarify this point.

The N=14 has been removed and the rest of the sentence has been clarified. 

• The objective two “determine if there is an association between plasma levels of polyamines and magnitude of pain behavior” is problematic. Results from animals treated with 14G2a (but not with DFMO) and controls showed no change at 24h and no earlier time points were analyzed as a proof of concept for this relation. In addition, the authors now indicate that a local change might take place (even during DFMO treatment) suggesting that plasma measurement could be an incorrect way to stablish a causal relation between polyamines and anti GD2 induced pain. These two arguments make difficult to support the speculation of such association, and the inclusion of both effects together, for example in the abstract “Administration of DFMO attenuated the enhanced sensitivity. Consistent with the known actions of DFMO on ornithine decarboxylase (ODC), serum putrescene and spermidine levels were significantly reduced by DFMO”, may give an incorrect idea of what the work does really demonstrate and confound the reader.

We stated the objective we set out to test. Allodynia and DFMO and their relationship with polyamine levels. 14G2a induced allodynia, an expected observation based on our published results and clinical observations. DFMO reduced polyamine levels. An expected finding. DFMO reduced allodynia. A novel finding and one of potential great significance and the determination of which was the PRIMARY objective of the study. However, despite DFMO reducing both PA and allodynia, their relationship was less than clear, requiring some speculation. One of those speculations is regarding local polyamine levels. This makes a lot of sense when one remembers the limited expression of GD2 to “…peripheral nerves, elements of the central nervous system, and skin melanocytes (2). (Page 4/31) In contrast, many cancer cells including neuroblastoma express GD2 on their surface (3).” (Page 4/31). Polyamines and ODC, in contrast, will be present in every cell, and serum polyamines abundant. DFMO will reduce both systemic and local levels of PAs. 14G2a, on the other hand, would be expected to only have actions at the GD2 antigen. Thus any increase in PA by anti-GD2 may be only at the local level and may not elevated to sufficient levels systemically and, in contrast to the decrease by DFMO, may not be detected in the serum by our assay. Since we don’t know if this is true or not, we added “local levels” as a very reasonable speculation.

To avoid any misunderstanding, we have modified the abstract to reflect that despite a significant decrease in anti-GD2 induced allodynia by DFMO, and despite a significant decrease in PA levels by DFMO, there lacked a conclusive relationship between the decrease in PA and the attenuation of anti-GD2 induced pain by DFMO “Abstract Results: …Consistent with the known actions of DFMO on ornithine decarboxylase (ODC), serum putrescene and spermidine levels were significantly reduced by DFMO, though the decrease in polyamine levels did not directly correlate with the behavioral changes.”

• In the new version, the results are even more complicated including statistics at different time points and concentrations, and some of them seem repeated in figure legends, including statistics. The authors may explore other forms of writings the results in a more clear and direct way and take advantage of visual support (figures-tables) to discharge this section. Consider to include it inside the manuscript, better than in supplementary material, since it is something relevant and helpful for the reader.

We appreciate the reviewer efforts to make the paper a better read. We have shortened the figure 2 and 3 legends substantially, and turned S1 Table into Table 1.

• After rewriting the discussion, these sentences “Spermine and spermidine induce a biphasic dose dependent activation/inactivation of NMDA receptors (NMDAR) on cortical neurons and induce pain behavior that can be blocked with NMDAR antagonists (33, 34). It is unknown whether these spinal actions are due to presynaptic or postsynaptic actions.” may not be correct. Neither of these works seem to look at the spinal cord. Ref 34 is a study at a peripheral level, and, if fact, shows that NMDAR are not implicated in the nociceptive actions of polyamines (as the authors indicate in fig 4). Perhaps the work by Kolhekar et al. 1994 (Neuroscience 63(4):925-36) could be more adequate to support their affirmation. But anyway the authors must be attentive to this.

We appreciate the reviewers’ insights in bringing this important work to our attention and have revised this section. 

• Minors:

In page 7/31 is written isoflorane instead of isoflurane

• In page 10/31 is written Prizm instead of Prism

Corrected.

• Reviewer #2: Authors reply to the observations. Please spell out all the abbreviations the first time they appear

• Ex in the abstract DFMO is spelled out the second time please correct.

Corrected.

---

## [Decision Letter · Decision Letter 2]

30 Jun 2020

Anti-GD2 induced allodynia in rats can be reduced by pretreatment with DFMO

PONE-D-20-03409R2

Dear Dr. Diccianni,

We’re pleased to inform you that your manuscript has been judged scientifically suitable for publication and will be formally accepted for publication once it meets all outstanding technical requirements.

Kind regards,

Giuseppe Biagini, MD

Academic Editor

PLOS ONE

Additional Editor Comments (optional):

Reviewers' comments:

Reviewer's Responses to Questions

**Comments to the Author**

1. If the authors have adequately addressed your comments raised in a previous round of review and you feel that this manuscript is now acceptable for publication, you may indicate that here to bypass the “Comments to the Author” section, enter your conflict of interest statement in the “Confidential to Editor” section, and submit your "Accept" recommendation.

Reviewer #1: All comments have been addressed

Reviewer #2: All comments have been addressed

2. Is the manuscript technically sound, and do the data support the conclusions?

Reviewer #1: Yes

Reviewer #2: Yes

3. Has the statistical analysis been performed appropriately and rigorously? 

Reviewer #1: N/A

Reviewer #2: Yes

4. Have the authors made all data underlying the findings in their manuscript fully available?

Reviewer #1: Yes

Reviewer #2: Yes

5. Is the manuscript presented in an intelligible fashion and written in standard English?

Reviewer #1: Yes

Reviewer #2: Yes

6. Review Comments to the Author

Reviewer #1: The authors are aware of the concerns indicated and have taken the actions that they have considered appropriate for their work. I have no further comments.

Reviewer #2: Authors reply is satisfactory.

Take into account that the resolution of the figures is rather bad, even downloading

them.

7. PLOS authors have the option to publish the peer review history of their article (what does this mean?). If published, this will include your full peer review and any attached files.

Reviewer #1: No

Reviewer #2: No

---

## [Editor Report · Acceptance letter]

6 Jul 2020

PONE-D-20-03409R2 

Anti-GD2 induced allodynia in rats can be reduced by pretreatment with DFMO 

Dear Dr. Diccianni:

I'm pleased to inform you that your manuscript has been deemed suitable for publication in PLOS ONE. Congratulations! Your manuscript is now with our production department. 

Kind regards, 

on behalf of

Dr. Giuseppe Biagini 

Academic Editor

PLOS ONE